

# Not just crop or forest: building an integrated land cover map for agricultural and natural areas

Melanie Kammerer[1,2], Aaron L. Iverson[3], Kevin Li[4], Sarah C. Goslee[1]

[1]USDA-ARS Pasture Systems and Watershed Management Research Unit, University Park, PA 16802, USA
[2]Oak Ridge Institute for Science and Education, Oak Ridge, TN 37860, USA
[3]Department of Environmental Studies, St. Lawrence University, Caton, NY, 13617, USA
[4]School for Environment and Sustainability, University of Michigan, Ann Arbor, MI, 48109, USA

*Correspondence to*: Melanie Kammerer (melanie.kammerer@usda.gov) or Sarah Goslee (sarah.goslee@usda.gov)

**Abstract.** Due to our increasing understanding of the role the surrounding landscape plays in ecological processes, a detailed characterization of land cover, including both agricultural and natural habitats, is ever more important for both researchers and conservation practitioners. Unfortunately, in the United States, different types of land cover data are split across thematic datasets that emphasize agricultural or natural vegetation, but not both. To address this data gap and reduce duplicative efforts in geospatial processing, we merged two major datasets, the LANDFIRE National Vegetation Classification (NVC) and USDA-NASS Cropland Data Layer (CDL), to produce an integrated land cover map. Our workflow leveraged strengths of the NVC and the CDL to produce detailed rasters comprising both agricultural and natural land-cover classes. We generated these maps for each year from 2012-2021 for the conterminous United States, quantified agreement between input layers and accuracy of our merged product and published the complete workflow necessary to update these data. In our validation analyses, we found that approximately 5.5% of NVC agricultural pixels conflicted with the CDL, but we resolved most of these conflicts based on surrounding agricultural land, leaving only 0.6% of agricultural pixels unresolved in our merged product. These ready-to-use rasters characterizing both agricultural and natural land cover will be widely useful in environmental research and management.

## 1. Introduction

Most agricultural landscapes include natural and semi-natural habitat, which supports biodiversity and ecosystem services critical for agricultural production (Cole et al., 2015; Cusser et al., 2019). Even small amounts of semi-natural habitat can provide disproportionate benefits for ecosystem services, including water quality, erosion mitigation, pest control, and pollination services (Bartual et al., 2019; Kremen and M'Gonigle, 2015; Zhou et al., 2014). Ecosystem services and the resulting benefits for agricultural production respond to both the amount and spatial configuration of agricultural and natural/semi-natural habitat patches. For example, pollinator communities vary based on area and proximity of semi-natural habitats (Kennedy et al., 2013), mass-flowering crops (Shaw et al., 2020) and even abundance of flowering weeds in a previous year's crop (Crochard et al., 2022). Conversely, agricultural habitats can have an inordinate impact on the ecological community and ecological functioning of surrounding natural areas. For instance, natural enemies, pathogens, or pesticides



from agricultural areas may spill over into bordering natural habitat, thus impacting the insect communities of those areas (Bell and Tylianakis, 2016; Long and Krupke, 2016; Rand et al., 2006).

35        Though spatial information on surrounding land cover is increasingly important for researchers and conservation practitioners, a single national dataset that includes both detailed natural and agricultural data does not exist. The Cropland Data Layer (CDL) is a widely used layer of land cover produced by the United States Department of Agriculture, National Agricultural Statistics Service (USDA NASS, 2021b), but, for many applications, the CDL lacks sufficient detail about semi-natural and natural habitats. The CDL classifies non-agricultural habitats based on the National Land Cover Dataset (Homer

et al., 2012), which has very broad classes of semi-natural and natural habitat. For example, for the entire United States, the CDL classifies forested areas as one of three forest classes (coniferous forest, deciduous forest, or mixed forest). These broad classes obscure known variance within a region in forest composition (Dyer, 2006) and conveys a false equivalence between distant geographic regions (e.g. 'mixed forest' in California bears little resemblance to 'mixed forest' in Maine, but they are a single class in the CDL). Similarly, information on crop identity is necessary for many ecological questions, but most land-

cover datasets that focus on natural and semi-natural habitats represent agricultural land in broad crop classes (e.g. 'row crop agriculture'), rather than specific crop types.

       Addressing this data gap, we combined two national datasets of land cover, the USDA-NASS Cropland Data Layer (CDL) and the LANDFIRE National Vegetation Classification (NVC), leveraging the strengths of both products (LANDFIRE, 2016c; USDA NASS, 2021b). The CDL delineates 110 agricultural classes, allowing researchers to quantify, for example,

how crop composition affects ecosystem services or biodiversity outcomes (e.g. mass-flowering canola provides abundant pollen and nectar for pollinators, while the same area of wheat has relatively few floral resources). Furthermore, because the CDL has been produced annually since 2008, researchers can examine temporal questions like changes in land cover or typical crop rotations in a specific area, although there are some caveats and best practices for these uses of the CDL (Lark et al., 2017). For instance, compared to reference data, the CDL historically undercounts cultivated area, but this bias has decreased

over time. To characterize change in land cover, analysts must adjust estimates of crop area to account for varying bias in the CDL (see recommendations in Lark et al., 2017).

       The National Vegetation Classification is a raster product produced by the Landscape Fire and Resource Management Planning Tools (LANDFIRE) program which specifies 537 vegetation classes, of which 420 are semi-natural or natural vegetation (LANDFIRE, 2016c) and 23 are agricultural. The NVC is released periodically, with the latest spatial product (LF

Remap, v2.0.0) corresponding to vegetation status in 2016. As the name suggests, the NVC raster is a spatial representation of vegetation classes defined by a standard vegetation classification recently developed for the United States (USNVC, 2016). Specifically, the LANDFIRE NVC maps USNVC vegetation at the group level, the sixth of eight levels of USNVC hierarchy. USNVC defines group as:

       A vegetation classification unit of intermediate rank (6th level) defined by combinations of relatively narrow sets of

65        diagnostic plant species (including dominants and co-dominants), broadly similar composition, and diagnostic growth



forms that reflect biogeographic differences in mesoclimate, geology, substrates, hydrology, and disturbance regimes (Federal Geographic Data Committee, Vegetation Subcommittee, 2008).

Compared with other national vegetation maps, LANDFIRE's National Vegetation Classification dataset confers several advantages to users. The USNVC publishes detailed descriptions of vegetation groups including typical geographic range, vegetation structure, dominant species, and a summary of climate, soils, and environmental factors (USNVC, 2016). In addition to spatial products, LANDFIRE distributes a national reference database of vegetation plots with each field plot labelled as an NVC vegetation class (LANDFIRE, 2016b). For some vegetation types, this reference database enables users to analyze plant communities of vegetation types included in the NVC raster, facilitating spatial representation of species distributions, plant functional traits, and ecosystem services. Also, the hierarchical structure of USNVC defines topological relationships between vegetation classes, allowing users to easily reclassify LANDFIRE products to coarser vegetation types (e.g. combine USNVC groups to map vegetation at the macrogroup or division level). Mapping coarser classes of land cover typically increases classification accuracy (Lark et al., 2021) and may be preferred for some applications.

To date, many environmental scientists have addressed the need for integrated layers of land-use by creating custom spatial datasets, which meet the needs of an immediate project but can lead to duplicated research effort and variability in final products. Geospatial data associated with specific research projects frequently cover small spatial extents, limiting reuse, and data are not commonly archived, with details of geospatial workflows being published with research results in discipline-specific journals, limiting findability. To address these challenges and reduce duplicative efforts in geospatial processing, we generated national datasets of agricultural and natural land-use from 2012-2021 and published the complete workflow necessary to update these data. We expect these pre-processed layers will be widely useful in environmental research, saving research effort by providing a ready-to-use raster characterizing detailed classes of both agricultural and natural land cover.

## 2. Methods

We developed a workflow to 1) merge a detailed map of natural vegetation (LANDFIRE National Vegetation Classification, NVC) with a detailed, annual map of agriculture (USDA-NASS Cropland Data Layer, CDL), 2) resolve conflicting land covers and 3) produce error statistics. We applied this workflow to the NVC v2.0.0 (also called '2016 Remap', LANDFIRE 2016a) and the CDL from 2012-2021 (USDA NASS, 2021b) for the conterminous United States. We reassigned agricultural pixels in the LANDFIRE National Vegetation Classification to a specific crop type that best matched their identity in the Cropland Data Layer. By altering only NVC agricultural classes (listed in Table 1), we ensured our merged vegetation map defined classes of non-agricultural vegetation that match the growth form specified in LANDFIRE vegetation height and cover layers. For example, pixels of 'Laurentian & Acadian Hardwood Forest' vegetation should correspond to pixels with tree metrics, rather than herbaceous or shrub, in the LANDFIRE layers for vegetation height and vegetation cover.



## 2.1 Geospatial workflow

For each of the 48 states in the conterminous United States, we buffered a vector layer of US state boundaries (U.S. Census Bureau, 2021) by 90m to accommodate edge effects of our raster processing. We clipped national CDL and NVC rasters to state boundaries and split each state raster into tiles of approximately 1000 km$^2$ with a 90m overlap between adjacent tiles. For each pair of CDL and NVC tiles, we executed the merge process, then joined output tiles together to generate state and national raster layers. We combined the NVC with CDL layers from 2012-2021 to create ten national annual rasters as our final output. We processed geospatial data by state to generate an archive of state rasters and facilitate joining land-use data with accuracy statistics from the CDL, which USDA-NASS publishes by state (see *Technical Validation* below).

We merged CDL and NVC tiles in two steps (Figure 1). First, for pixels specified as agricultural land in NVC and CDL layers, we reassigned the NVC agricultural class to a more specific crop type from the CDL (see Table 1 for NVC/CDL class matches). We defined NVC/CDL matches based on crop classes that are synonyms (e.g. NVC wheat = CDL wheat), subsets (e.g. NVC orchard = CDL apple, peach, cherry, etc.), or potential components of a crop rotation. For example, we considered NVC row crop as a match for all annual crops (e.g. corn, soybeans, wheat, vegetables) and perennial vegetation that can be rotated with annual crops (e.g. alfalfa, hay, or pasture). In some areas of the western United States, including fallow years in crop rotations is a common strategy for moisture conservation (Hansen et al., 2012), so we considered NVC 'Fallow-Idle Cropland' as a match to most CDL crops (Table 1). It was necessary to consider mismatch due to crop rotations because we combined the single 2016 NVC raster with CDL data from 2012-2021, where a single CDL pixel could have multiple crop designations across the timespan.

The second step of our workflow was reassigning NVC agricultural pixels were not agriculture in the CDL (or the two agricultural class identities did not match according to Table 1). For these pixels, we identified the most likely crop type based on surrounding agricultural pixels. Specifically, we calculated the dominant CDL crop type within 90 m (7 x 7 pixel neighborhood) and assumed the focal pixel was the same crop. To account for edge effects and ensure that the dominant crop type was accurately identified from a complete 7x7 neighborhood, we created raster tiles that overlap by 90 m (see above). We executed the merge procedure on overlapping tiles, then clipped each tile to remove the overlapping area. If there were no agricultural pixels within 90m of a mismatched pixel, we assigned that focal cell a value of -1001 indicating unresolved land cover due to mismatch between input rasters. In the technical validation (see below), we summarized the frequency of initial pixel mismatch between CDL & NVC input rasters (after workflow step one) and unresolved pixels remaining in merged raster (after workflow step 2).

We executed all geospatial operations in R v4.1.2 (R Core Team, 2021) using *terra* v1.5-22 (Hijmans, 2021), *raster* v3.5-15 (Hijmans, 2022), *SpaDES.tools* v0.3.10 (Chubaty and McIntire, 2022), and *gdalUtils* v2.0.3.2 (Greenberg and Mattiuzzi, 2020) within a Singularity container (available with code archive, Kammerer, 2022). To facilitate processing large geographic extents, we utilized a high-performance computing cluster via the USDA-ARS SCINet Initiative.



## 2.2 Technical Validation

We validated merged raster products to verify that our geospatial workflow produced the desired output and quantified spatial agreement between source layers. To ensure valid output, for all merged rasters, we verified that file size, spatial extent, coordinate reference system, and pixel values matched expected values (see data file x for attribute table for merged raster). All rasters in the linked archive (cite data archive) passed our automated testing regime.

We quantified spatial agreement between the NVC and the CDL by calculating the number and proportion of mismatched

pixels present after step one of our geospatial workflow (Figure 1). We defined mismatched pixels as NVC agricultural pixels that were not a relevant agricultural class in the CDL (Table 1). This included CDL pixels that were not agricultural (developed or natural land) or agricultural pixels that we deemed in conflict with the NVC agricultural class (see 'Geospatial workflow' above for rationale of class matching). After we re-assigned values for mismatched pixels (workflow step 2), we also summarized the number and proportion of pixels with unresolved land cover in our output rasters. From preliminary analyses,

we determined that there was relatively little variation in mismatched pixels over time (Figure S1, Figure S2), so we presented results for one representative year of the CDL, 2017 (Figure 2).

We assessed error of our merged data product by integrating accuracy statistics for the CDL and the NVC (LANDFIRE, 2016a; USDA NASS, 2021a). For each year and CDL class, USDA-NASS defines accuracy as the match between CDL raster and a reference dataset of administrative crop data and the National Land Cover Dataset (Homer et al.,

2012), another remotely sensed map of land cover. For each NVC region, LANDFIRE quantifies agreement between NVC raster and field plots in the LANDFIRE reference database (LANDFIRE, 2016b) labelled as specific NVC vegetation types. Due to error associated with labelling LANDFIRE field plots, 'agreement' rather than 'accuracy' is a more appropriate term for NVC assessment (LANDFIRE, 2016a). But, to utilize consistent language for both datasets, we referred to agreement between spatial and reference data as 'accuracy'. To summarize accuracy of CDL, NVC, and merged rasters, we calculated an

area-weighted mean of accuracy data per county in the conterminous United States.

To aid interpretation of our accuracy maps, we also generated county-level maps of the proportional area of land cover classes with ground-truth data (reference data coverage) for CDL, NVC, and merged raster layers. NVC accuracy data are only available for vegetation types with at least 30 field-surveyed plots in the LANDFIRE reference database (LANDFIRE, 2016b), and in some regions, many NVC classes were lacking accuracy values due to insufficient field plots. For the CDL and

NVC, we calculated coverage of reference data considering only agricultural and unmanaged classes, respectively, while coverage values for the merged product included agricultural and unmanaged classes. Like our assessment of pixel mismatch, we showed data on accuracy and coverage of reference data using the 2017 CDL.

## 3. Results

In our technical validation, we found that 5.5% of NVC agricultural pixels conflicted with the CDL (i.e., the CDL

was not an agricultural class or two agricultural class identities did not match according to Table 1). NVC classes for





pasture/hayland, and row crop had the highest number of conflicting pixels with CDL, with fewer mismatches for less common agricultural classes such as vineyard, orchard, fallow/idle, and aquaculture (Figure 2). Like other studies (Goslee, 2011; Lark et al., 2021), we found classification of pasture/hayland was a challenge, with 12.4% of NVC pixels of pasture/hayland conflicting with the 2017 CDL. For other common agricultural classes in the NVC (classes totaling more than 5% of U.S.

agricultural land), pixels in conflict with the CDL did not exceed 5.1% of class area (Figure S3). Agricultural classes that were less common in the NVC had fewer mismatched pixels but higher proportional mismatch with CDL. For example, 65% of aquaculture pixels in the NVC were not classified as aquaculture in the CDL, although the approximately 94,000 ha of NVC aquaculture represents only 0.05% of agricultural land in the United States. Greater than 18% of bush fruit and berries, orchard, vineyard, and aquaculture area conflicted with CDL, although total area of these classes only represents approximately 1.8%

of U.S. agricultural land.

Mismatch between NVC and CDL land cover represented a median of 6.4% of county area, although mismatch was more prevalent in the Western United States. Most U.S. counties had low to moderate mismatch with less than 13% and 24% area mismatch in 77% and 90% of counties, respectively. Counties in the Western U.S. had higher mismatch, with some counties in the Southwest, California, Minnesota, Idaho, and Montana exceeding 65% of agricultural land in the NVC

conflicting with the CDL. In counties with >65% mismatch, the most common mismatches were pixels classified as pasture/hayland or orchard in the NVC and shrubland in the CDL.

By reassigning cells based on the most common agricultural land in the surrounding area, we resolved most mismatched pixels. In our final raster product, most U.S. counties had very little area of unresolved pixels, but southern California, Texas, Louisiana along the Gulf of Mexico, and Florida were hotspots of conflicting pixels with no agricultural

land within 90m. In 96% of U.S. counties, unresolved pixels in the final raster were less than 0.4% of county area. Excepting one county in Louisiana, in hotspot areas, pixels with unresolved land cover represented 0.4-4.6% of county area. The most common cause of unresolved pixels was conflict related to classifying pasture/hayland, shrublands, wetlands, vineyard, and tree fruits. For example, in hotspots of unresolved pixels in Texas and Florida, pixels that the NVC classified as pasture/hayland were shrubland, wetland, or forest classes in the CDL. We could not resolve conflicts in these areas because there was no

agricultural land within the 90m search radius due to high proportion of non-agricultural land. In some counties in southern California, up to 4.6% of county area was unresolved land cover because 1) NVC orchard or vineyard pixels were assigned to grass/pasture or shrubland in the CDL or 2) NVC row crop or vineyard pixels were tree nuts (almonds, walnuts, or pistachios) in the CDL. When conflicting land cover in California included orchard or vineyard classes, unresolved pixels typically represented a whole field (large, regular shape). If users are focusing on counties with more unresolved pixels, we recommend

collecting ground-truth data or consulting additional imagery or land cover data to resolve mismatch between CDL and NVC layers.

The LANDFIRE program uses the CDL to classify broad agricultural types in the NVC, and one aspect of the LANDFIRE workflow likely contributed to mismatched pixels between the NVC and the CDL. In previous years, CDL models



sometimes generated noise pixels scattered across a field that were misclassified as other crops. To address this, LANDFIRE
executed a zonal smooth reclassifying CDL to the majority crop type within polygons of ownership. Then, LANDFIRE utilized
the smoothed CDL to define agricultural classes for NVC (LANDFIRE program, personal communication). We could not
replicate the LANDFIRE methodology because the spatial layer of ownership polygons (Common Land Unit) is not publicly
available (USDA, 2017) but, when CDL and NVC conflicted, we also determined crop identity from surrounding crop pixels.
Also, in recent years, USDA-NASS improved within-field classification of the CDL and, starting with a 2020 update to
LANDFIRE products depicting fuels, fire, and vegetation height/cover, LANDFIRE dropped the zonal smooth from their
workflow (LANDFIRE program, personal communication).

At a national scale, classification accuracy of the CDL was higher than the NVC and accuracy of our merged product
was between the CDL and NVC values because we calculated accuracy of the merged product as a weighted average of
accuracy of the input layers. For CDL, NVC, and the merged product, we calculated accuracy per county yielding a distribution
of accuracy values (Figure S4). For user's and producer's accuracy of the 2017 CDL, the median of county accuracy values
was 75.8% and 81.5%, respectively, compared with 49.5% and 51% for the NVC (Figure S4). For our merged product, median
user's accuracy was 57% and producer's accuracy was 56.7%. Accuracy of our merged dataset fell between 2017 CDL and
NVC values (Figure S4), albeit closer to the NVC due to high proportion of natural land in many counties.

Accuracy of the CDL, NVC, and our merged product varied geographically. The 2017 CDL was most accurate in
high agricultural areas, especially the Great Plains and much of the Midwestern Corn Belt, with lower accuracy in the Mid-
Atlantic, Southeast, Upper Midwest, Pacific Northwest, and Southwest regions (Figure 3). For the NVC, we did not find any
relationship between accuracy and the amount of natural land in each county, likely because large areas of natural land can
comprise many habitat types, while large swaths of agriculture are typically simplified landscapes with relatively few crop
types. We found a regional hotspot of user's accuracy exceeding 72% from Iowa and western Missouri south to central Texas
(Figure 3). User's and producer's accuracy of the NVC were both low (< 37%) in Florida, eastern Missouri, and some areas
in Ohio and Indiana. In regions including Florida, western Texas, and Missouri and Arkansas, the LANDFIRE program had
very little reference data and was not able to assess accuracy of many vegetation types (see coverage of reference data below).
Even for vegetation types included in the accuracy assessment, the resulting accuracy values were likely lower due to the lack
of reference data from these areas. Accuracy of our merged product most closely resembled the NVC, with higher accuracy in
the Midwestern Corn Belt due to prevalence of agriculture in this region (Figure 3). It would be interesting to analyse
relationships between environmental and vegetation characteristics and classification accuracy, but this was beyond the scope
of the current work.

We found that coverage of reference data for the CDL was consistently high, but, for the NVC (and therefore our
merged products), coverage ranged from 1.3% to nearly 100% by county (Figure 4, Figure S5). For CDL, coverage of reference
data was >93% for the whole conterminous United States because the CDL accuracy assessment included nearly all crop
classes. For the NVC and our merged raster, median coverage was 77.2% and 84.4%, respectively. Most counties with NVC
reference coverage less than approximately 50% were in the Midwest, Texas, Florida, and coastal areas along the Gulf of

Mexico and Atlantic Ocean (Figure 4). In Florida, western Texas, and Missouri and Arkansas, reference coverage for the NVC was low in many counties that had moderate to high proportional area of natural vegetation (Figure S6). This suggests that

even relatively abundant vegetation types in these areas had insufficient reference data to be included in LANDFIRE's accuracy assessment. Lack of reference data in these areas likely also contributed to low accuracy values. For the NVC, reference coverage less than approximately 30% corresponded to lower accuracy values (Figure S7). Like our accuracy results, for our merged product, spatial patterns of reference coverage resembled NVC with higher values in the Midwestern Corn Belt (Figure 4).

**4. Usage Notes**

We created a new spatial dataset that integrates both agricultural and natural land cover and that can be used by environmental scientists and managers who are interested landscape-scale processes involving both land cover categories. Though we were able to reduce additional inaccuracies from artifacts of our geospatial processing, the product we created is still subject to limitations of the source data. Accuracy of CDL is highest in regions dominated by agriculture and lower in

states with more developed or semi-natural land cover, particularly mixed-use landscapes (USDA NASS, 2021a; Lark et al., 2021). LANDFIRE vegetation products like the NVC are designed to be used at landscape, regional, or national scales and are likely less accurate for very small spatial extents (LANDFIRE, 2016a). For rare vegetation classes, LANDFIRE does not publish data on agreement between field plots and NVC raster. If projects are targeting small spatial extents or rare vegetation types, we recommend users review our land cover maps and adjust based on local knowledge or additional vegetation surveys.

**Data Availability**

All data files generated in this work are archived with the USDA Ag Data Commons as follows:

1.  Tabular data from technical validation, county accuracy statistics, and an attribute table for merged rasters are available at https://doi.org/10.15482/USDA.ADC/1527977 (Kammerer et al., 2022b).
2.  National merged rasters from 2012-2021 are available at https://doi.org/10.15482/USDA.ADC/1527978 (Kammerer

et al., 2022a).

**Code Availability**

To facilitate updating these spatial layers with new versions of LANDFIRE NVC or additional years of CD, we archived all code to generate merged rasters on Zenodo at https://doi.org/10.5281/zenodo.6803199 (Kammerer, 2022).

**Author contribution**

All authors conceived and designed the study. MK developed code for geospatial analyses and generated validation data. All

authors contributed to data visualization and interpretation. MK drafted the manuscript and all authors contributed critically

to the drafts and gave final approval for publication.

**Competing Interests**

The authors declare that they have no conflict of interest.

**Acknowledgements**

We are grateful to Dr. Debra Peters and the Peters research group for feedback on this project. We also thank the staff

at the USDA National Agricultural Library for assistance with data archiving. This project was funded by the SCINet project

of the USDA Agricultural Research Service, ARS project number 0500-00093-001-00-D.

This research was supported in part by the U.S. Department of Agriculture, Agricultural Research Service. The

findings and conclusions in this manuscript are those of the author(s) and should not be construed to represent any official

USDA or U.S. Government determination or policy. Mention of trade names or commercial products in this publication is

solely for the purpose of providing specific information and does not imply recommendation or endorsement by the USDA.

USDA is an equal opportunity provider and employer.

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



**Figures**

**Figure 1: Geospatial workflow to combine USDA-NASS Cropland Data Layer (CDL) and LANDFIRE National Vegetation Classification (NVC) rasters. Panel A shows input raster layers. Panel B shows results of reclassifying NVC agricultural pixels to CDL values. Mismatched pixels in Panel B were the result of conflicts between the NVC and CDL (see Table 1 for class matches). We resolved pixel conflicts by assigning mismatched pixels to the most common CDL agricultural class within a 3 pixel radius (Panel C). When there were**
**no proximate CDL agricultural pixels, we could not resolve mismatch, leaving some unresolved pixels in our final rasters (Panel C). Top and bottom rows are, respectively, all classes of land cover and land cover reclassified to binary agriculture or non-agriculture.**



**Figure 2: Frequency of disagreement between LANDFIRE National Vegetation Classification (NVC) agricultural classes and 2017 Cropland Data Layer (CDL). Panels A and B depict frequency of pixel disagreement in the original land-cover rasters (after step 1 of our geospatial processing, Figure 1). Panel A shows specific CDL classes that conflicted with NVC agricultural classes and, for each county in the conterminous United States, Panel B depicts the percentage of NVC agricultural pixels that did not match CDL. For the final merged raster (output of workflow step 2), Panel C shows the percentage of each county where conflict between CDL and NVC layers could not be resolved. To facilitate mapping, we converted percentages to discrete intervals using Jenks natural breaks algorithm.**



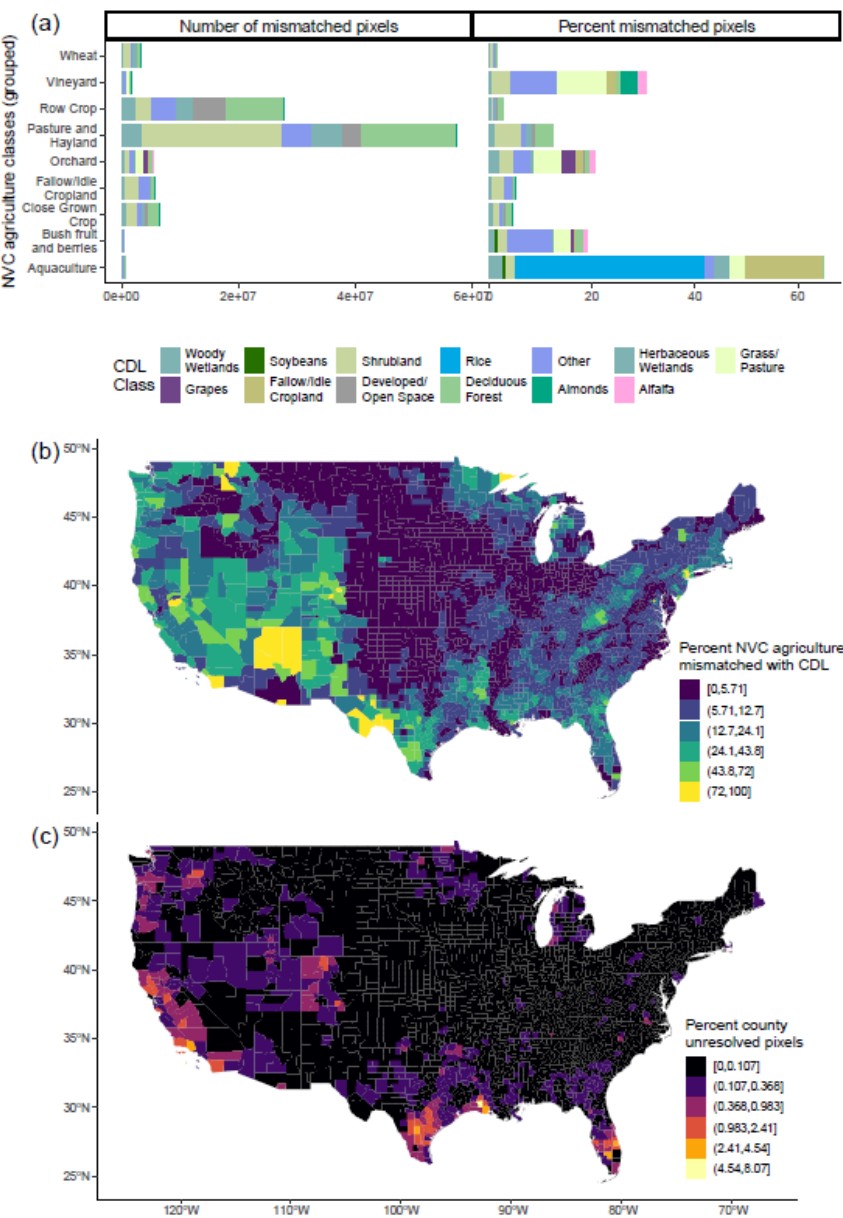



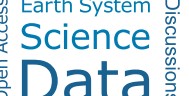

**Figure 3: For 2017 CDL, NVC, and new merged product (CDL+NVC), accuracy of land cover classification compared against reference data. Accuracy statistics were calculated per county as an area-weighted mean of accuracy values assigned to each land cover class. For visualization purposes, we converted accuracy values to discrete intervals using Jenks natural breaks algorithm.**



**Figure 4. Percentage of county area with reference data for the conterminous United States. For 2017 CDL, reference data were from a USDA administrative crop database and National Land Cover Dataset imagery. For NVC, reference data were field plots in LANDFIRE reference database, with a minimum of 30 field plots per NVC class. For CDL and NVC, we calculated coverage of reference data as a fraction of agricultural and unmanaged classes, respectively, while, for the merged product, coverage of reference data included agricultural and unmanaged classes. To facilitate mapping, we converted values to discrete intervals using Jenks**
**natural breaks.**

$$\text{Coverage of reference data}_{\text{CDL}} = \frac{\text{Area}_{\text{ag with ground-truth}}}{\text{Area}_{\text{all ag}}}$$

Legend:
- [1.29,26.1]
- (26.1,46.7]
- (46.7,64.8]
- (64.8,80.5]
- (80.5,93.4]
- (93.4,100]

$$\text{Coverage of reference data}_{\text{NVC}} = \frac{\text{Area}_{\text{unmanaged in RefDB}}}{\text{Area}_{\text{all unmanaged}}}$$

$$\text{Coverage of reference data}_{\text{NVC+CDL}} = \frac{\text{Area}_{\text{ag with ground-truth}} + \text{Area}_{\text{unmanaged in RefDB}}}{\text{Area}_{\text{all ag}} + \text{Area}_{\text{all unmanaged}}}$$



## Tables

**Table 1: Agricultural land cover classes in LANDFIRE National Vegetation Classification (NVC) we considered a match to the indicated classes in USDA-NASS Cropland Data Layer (CDL).**

| NVC Name | NVC Value(s) | Name(s) of Matching CDL Classes | Value(s) of Matching CDL Classes |
|---|---|---|---|
| Vineyard | 7961, 7971, 7981, 7991 | Grapes | 69 |
| Bush fruit and berries | 7962, 7972, 7982, 7992 | Blueberries, Cranberries | 242, 250 |
| Orchard | 7960, 7970, 7980, 7990 | Almonds, Apples, Apricots, Avocados, Cherries, Christmas Trees, Citrus, Nectarines, Olives, Oranges, Other Tree Crops, Peaches, Pears, Pecans, Pistachios, Plums, Pomegranates, Prunes, Walnuts | 75, 68, 223, 215, 66, 70, 72, 218, 211, 212, 71, 67, 77, 217, 210, 76 |
| Aquaculture | 7979, 7989, 7999 | Aquaculture, Open Water | 92, 111 |
| Row Crop - Close Grown Crop | 7963, 7973, 7983, 7993 | Fallow/Idle Cropland, Grass/Pasture, Other Hay/Non Alfalfa, All Other Crops* | 61, 176, 37, various |
| Row Crop | 7964, 7974, 7984, 7994 | | |
| Close Grown Crop | 7965, 7975, 7985, 7995 | | |
| Fallow/Idle | 7966, 7976, 7986, 7996 | | |
| Pasture and hayland | 7967, 7977, 7987, 7997 | | |
| Wheat | 7968, 7978, 7988, 7998 | | |


*All Other Crops = Alfalfa, Asparagus, Avocados, Barley, Broccoli, Buckwheat, Cabbage, Camelina, Caneberries, Canola, Cantaloupes, Carrots, Cauliflower, Celery, Chick Peas, Clover/Wildflowers, Corn, Cotton, Cucumbers, Dbl Crop Barley/Corn, Dbl Crop Barley/Sorghum, Dbl Crop Barley/Soybeans, Dbl Crop Corn/Soybeans, Dbl Crop Durum Wht/Sorghum, Dbl Crop Lettuce/Barley, Dbl Crop Lettuce/Cantaloupe, Dbl Crop Lettuce/Cotton, Dbl Crop Lettuce/Durum Wht, Dbl Crop Oats/Corn, Dbl Crop Soybeans/Cotton, Dbl Crop Soybeans/Oats, Dbl Crop Triticale/Corn, Dbl Crop WinWht/Corn, Dbl
Crop WinWht/Cotton, Dbl Crop WinWht/Sorghum, Dbl Crop WinWht/Soybeans, Dry Beans, Durum Wheat, Eggplants, Flaxseed, Garlic, Gourds, Greens, Herbs, Honeydew Melons, Hops, Lentils, Lettuce, Millet, Mint, Misc Vegs & Fruits, Mustard, Oats, Onions, Other Crops, Other Small Grains, Peanuts, Peas, Peppers, Pop or Orn Corn, Potatoes, Pumpkins, Radishes, Rape Seed, Rice, Rye, Safflower, Sod/Grass Seed, Sorghum, Soybeans, Speltz, Spring Wheat, Squash, Strawberries, Sugarbeets, Sugarcane, Sunflower, Sweet Corn, Sweet Potatoes, Switchgrass, Tobacco, Tomatoes, Triticale, Turnips, Vetch, Watermelons, Winter Wheat
