# Peer review of "Figure S1: Frequency of disagreement between LANDFIRE National Vegetation Classification (NVC) agricultural classes and 2012-2021 Cropland Data Layer (CDL). These data are frequency of pixel disagreement in the original land-cover rasters (after step 1 of our geospatial processing, Figure 1). For si"

_Earth System Science Data, 2022_

## Author Comment (AC2)

Anonymous Referee #2, 27 Jan 2023

We thank the reviewer for a thoughtful and thorough review of our manuscript and provide line-by-line responses in blue text.

This manuscript describes a data integration effort, combining two existing vegetation/cropland cover datasets for the conterminous United States, namely the LANDFIRE National Vegetation Classification (NVC) and USDA-NASS Cropland Data Layer (CDL). As I am not an expert in the application domain for these datasets, I will focus on the integration process itself and on the validation of the resulting data product.

I think that the concept underlying this effort ("generating new knowledge by integrating existing datasets") is very valid and timely – this is how we need to deal with the myriad of geospatial data out there in order to get the maximum value out of it. The paper is well written, and from my perspective, such an effort is of potential interest for the readers of ESSD, however, there are several concerns that should be addressed prior to publication. My biggest concern is the temporal mismatch of the data (i.e., vegetation data from 2016 is integrated with agricultural data at annual temporal resolution from 2012-2020) – why not using all available Landfire epochs, or constraining the dataset to 2016 only? This temporal mismatch should be addressed in the revised version and the rationale for this decision should be clarified.

Thank you for your comments about the value and timeliness of our work. We agree that data integration is a key aspect of gaining insight from large (and increasing) volume of available geospatial data. In response to the reviewer's questions and feedback about the temporal mismatch of CDL vs. NVC, please see our response under #4 below. We appreciate the reviewer bringing up this point and agree that it required additional clarification and justification.

Specific comments:

1. The benefit / underlying motivation of this data integration effort needs to be clarified a little bit more. While the introduction refers to some examples in the literature that assess processes playing out at the interface of cropland and natural vegetation, the Authors should add a paragraph (maybe in a concluding section) to illustrate some technical examples how these data could be used to answer specific questions – for example, one could apply convolving focal windows to identify regions where specific crop / vegetation types co-occur within a given distance. Something like this would make the contribution/value of the integrated dataset clearer.
   We agree that providing more examples would be beneficial, and we thank reviewer 2 for contributing a specific use case to examples we provided. In response to this comment and feedback from reviewer one, we added the following text to the introduction to provide examples of how SPAN could be used. We felt that including it earlier in the manuscript, rather than in the conclusion, would better contextualize the product early on.
   "We envision many diverse uses cases for SPAN, including predicting biodiversity, ecosystem services, and climate adaption and mitigation strategies. For example, The Integrated Valuation of Ecosystem Services and Tradeoffs (InVEST) is a set of widely used, spatial models that predict

ecosystem services based on land cover data. InVEST predictions of crop pollination services depend on accurate characterization of agricultural and natural habitats available in one spatial product. Within broad classes of agricultural, forest, and wetland habitats, floral resources for pollinators can vary more than 250, 750, and 40-fold, respectively (Iverson et al in prep), necessitating a land cover map that specifies specific types of crop and natural vegetation. We developed SPAN as an input for models of pollination services, but expect models of carbon storage, crop disease, pest dynamics, and biocontrol, among other ecosystem services, would be improved with more detailed land cover data. We also foresee a variety of non-modelling uses for SPAN, such as topological analyses to identify proximity of specific crop and vegetation types."

2. Also, a vectorized version (polygons) of the integrated dataset could be very useful to assess topological relationships (e.g. adjacency) between different crop / vegetation types). à If feasible, the Authors could provide such a vectorized polygonal dataset to complement their data. This will also enhance the usage of the dataset, as some researchers may prefer to work with vector rather than raster data, for topological analyses, but may not have the resources to vectorize the data.

   We appreciate this suggestion from the reviewer and agree that vector data could be useful to some users. To assess the feasibility of providing SPAN in a polygon format, we vectorized one county of the 2021 SPAN raster. We found that converting to a vector format (.gpkg) increased file size by 266% relative to raster format. We estimate that, in vector format, one year of SPAN would be a 10.6GB file and all ten years over 100GB. While some researchers may prefer to work with vector data, we judged that, for most users, working with even a single year of vectorized SPAN at >10GB would be very difficult (slow to download and nearly impossible to manipulate on typical laptop or desktop computers). Given these challenges with working with the vector data, we decided to provide SPAN solely in raster format.

3. The validation section should be expanded. While some sort of validation has been done, little information is given about the reference data used – please provide some information (maybe a map) on the sampling locations, data source of the reference data etc. –

   In the third paragraph under 'Technical Validation,' we provide readers with the data source of the reference data for CDL and NVC (unpublished dataset of crop acreage and LANDFIRE reference dataset, respectively). For CDL, the size of validation dataset and locations where these data were collected are not publicly available from NASS, so we are unable to add this information to our manuscript.

   Though we would be willing to include a detailed description of the NVC ground truth data if the reviewers/editor feel we should, we believe it does not fit within the scope of our paper. To clarify our objectives and guide readers who may be interested in these data, we added the following text to the methods section.

   "We do not include a detailed description of CDL and NVC reference data because our goal was to illustrate how the classification accuracy of SPAN compares to accuracy of CDL and NVC, rather than validating the source datasets themselves. For more information on CDL and NVC validation procedures, we refer readers to CDL and LANDFIRE websites (https://www.nass.usda.gov/Research_and_Science/Cropland/sarsfaqs2.php#Section1_11.0 and https://landfire.gov/remapevt_assessment.php, respectively) and Lark et al. (2021).

4. Only after one hour of reviewing this paper I realized the temporal mismatch in the data, when reading this sentence "Pixels of national vegetation are the same in all rasters provided here and represent land use in 2016." on the data website (https://data.nal.usda.gov/dataset/data-not-just-crop-or-forest-building-integrated-land-cover-map-agricultural-and-natural-areas-spatial-files/resource/8c92879b-92cf-4e86-a3c4-0e672007a1df) - NVC data from 2016 is integrated with CDL data annually from 2012-2020? This is not clear from the manuscript. What are the implications of keeping vegetation cover stationary over time? Does cropland change faster than vegetation? How does this temporal mismatch affect the usability of the integrated dataset? Can it be used to assess recent processes at all? This issue needs to be highlighted and thoroughly discussed. When looking at this page: https://landfire.gov/data_overviews.php, I see that Landfire has been released in several years besides 2016 – why did you not integrate Landfire and CDL in annual pairs, for the years available? Please provide a rationale for this. This is probably my biggest concern about this manuscript.

We appreciate this helpful feedback and insightful questions from the reviewer. To explain why we used only 2016 NVC, we added the following text to the introduction and methods.

"Currently, the only available NVC product represents vegetation status in 2016 (LF Remap, v2.0.0), but LANDFIRE has indicated plans to release additional, updated versions (LANDFIRE program personnel, personal communication)."
"We integrated 2012-2021 CDL with the NVC from 2016 because, at present, 2016 is the only year with an NVC raster. Recent releases from LANDFIRE (LF 2019 & 2020) updated maps of vegetation attributes (e.g. vegetation cover and height), but did not affect the vegetation type products, including NVC. LANDFIRE also distributes an existing vegetation type ('EVT') product for 2001 and 2016, but we used NVC because it corresponds to a vetted schema classifying vegetation types (USNVC 2016). Also, 2008 is the first year of the national CDL, so including the 2001 EVT vegetation map would add little value over only using a 2016 vegetation map. If future LANDFIRE releases include additional NVC maps, we anticipate updating natural vegetation in SPAN."

We addressed the reviewer's questions about the implications of keeping vegetation cover stationary by adding this paragraph to the Usage Notes:
"SPAN is based on a static representation of natural vegetation circa 2016, so likely contains errors for vegetation types or geographic regions that are rapidly changing or experience frequent disturbance. By including CDL for 2012-2021, we captured the dynamic nature of agricultural land cover due to crop rotations, but the total amount of natural vs. agricultural land is fixed based on land cover in 2016. Consequently, SPAN is not appropriate for analyses of natural land conversion to agriculture or vice versa. With future releases of the NVC, we hope to more frequently update natural vegetation in SPAN to better align with other common LULC products (e.g. 5-year release schedule of National Land Cover Database, Homer et al 2012)."

5. Same sentence on the data website : "Pixels of national vegetation are the same in all rasters provided here and represent land use in 2016." --> isn't vegetation land cover, instead of land use?

Per the reviewer's suggestion, we changed this sentence to indicate land cover instead of land use.

6. If not done already, Authors should provide a spatial layer (raster dataset) of the mismatched / unresolved pixels, so that users can include these discrepant areas explicitly in their analyses. Unresolved pixels are already included in the SPAN final rasters and corresponding attribute table (class number -1001). We agree that some users may be interested in specific location of mismatched pixels, so, for each year, we created a raster layer depicting this information. We added these rasters to the data repository and the following reference in the manuscript text: "For users interested in specific locations of mismatched pixels, in the accompanying data archive (Kammerer et al., 2022a), we provided annual raster files identifying all mismatched pixels."

7. Related to that, it is not clear to me in which year the agreement assessment was conducted (sorry if I missed it), and whether the stats (e.g. 5% of conflicting pixels) refers to a specific year, is it 2016?
We included results for all years (2012-2021) in the Supplemental Materials (Figures S1 and S2), but, in the main manuscript, presented results only for 2017. We edited the methods (2nd paragraph of 'Technical validation') to clarify that we examined agreement data for all years. In the methods and captions to figures 2 and 3, we previously specified that agreement statistics were for 2017. In the updated manuscript, we also mention the 2017 date at the start of the relevant results paragraphs. From preliminary analyses, we determined that there was relatively little variation in mismatched pixels over time (Figure S1, Figure S2), so we presented results for one representative year of the CDL, 2017.

8. It seems that the union of agricultural land use and vegetation land cover is used as the analytical "universe" in this study. How do these areas relate to other land cover / land use types, such as urban areas / developed land? It would be great if the Authors could conduct a cross-comparison to a "spatially exhaustive" dataset such as the NLCD – how does the integrated dataset agree with the classes from NLCD? This would be some kind of "external" evaluation, while the agreement analysis would be an "internal" validation. Perhaps a cross-tabulation of the area proportions per crop/vegetation class and NLCD land cover class would be interesting (see e.g. Fig 4 in https://www.nature.com/articles/s41597-022-01591-0)
Thank you for these questions and feedback, which we addressed by adding the following text to the introduction:
"In this work we focused on agricultural land use and natural land cover, as these are the primary strengths of the CDL and NVC datasets, respectively. But, based on the National Land Cover Dataset, the NVC raster includes four classes of urban/developed land. We retained these classes in SPAN, enabling users to explore questions like proximity or diversity of natural vegetation near urban areas."
Also, regarding the reviewer's suggestion to compare SPAN with the NLCD, because the NLCD was used to define urban areas in the NVC, and consequently, SPAN, we could not also use NLCD as a dataset for external evaluation.

9. I suggest to rename the dataset containing the uncertainty statistics. Please change "tabular data" to "uncertainty statistics" or similar.

We elected not to rename this data resource because, in addition to the uncertainty statistics, it contains the tabular attribute table for the merged rasters. Due to procedures of the USDA Ag Data Commons, we could not include the raster attribute table with the spatial files in the same dataset. We agree with the reviewer that it would be clearer to separate spatial data (and associated files) from uncertainty information and name accordingly, but we were stymied by bureaucracy.

10. Lastly, I suggest to come up with a name for the integrated dataset. This will make it easier to refer to the dataset and ultimately, increase the visibility of the product.

We appreciate this helpful feedback to increase visibility of the product. We decided to call the dataset 'Spatial Products for Agriculture and Nature' (SPAN) and incorporated the name throughout the manuscript.

Minor comments:

1. Sorry if I missed it, but what is the spatial resolution of the input data? I think it is 30m for both of the datasets, this should be stated in your geoprocessing section. Also, it is important to know whether the raster grids align, or is there an offset between the two grids? Did you have to resample one of the layers to the grid of the other layer? If so, how was this done (nearest neighbor resampling?).

Thank you for this comment. We clarified these aspects by adding the following text to the methods section:

"The NVC and CDL have a 30m spatial resolution and were available from USDA NASS and LANDFIRE, respectively, in the NAD83 / Conus Albers projection (EPSG:5070). In their native format, NVC and CDL grids aligned, so we did not resample either dataset. "

2. Some terminology… AFAIK one would speak of "forest land cover" on the one hand, but "agricultural land use" on the other hand – in the title and in the manuscript you write "land cover" – the term "land use" does not occur in the manuscript. However, isn't your integrated dataset truly a LULC (land use / land cover) dataset? I think the integration of land cover and land use should be highlighted more in the paper (and maybe even in the title).

The reviewer makes a good point, and we recognize there are varying definitions for land use/land cover in the literature (Nedd et al. 2021). To clarify terminology, we added the following text to the manuscript introduction: "According to Verburg et al (2011), land cover is "the layer of soils and biomass, including natural vegetation, crops and human structures that cover the land surface." We used this definition and considered specific crop types delineated by the CDL to be land cover, rather than land use."

3. Minor detail: Some of the maps in Figures 2,3,4 show areas / area proportions and thus, should use an equal-area projection rather than showing Lat-Lon in a cartesian coordinate system. Lat-Lon are angular coordinates and should IMHO not be shown in cartesian coordinate systems, in particular when it comes to mapping areas / densities / area proportions – as a pixel in Maine has a different area than a pixel in Florida. I suggest to use Albers Equal Area projection.

We appreciate the reviewer flagging this issue. As recommended, we changed maps depicting area or proportions (Figures 2-4 and supplemental material) to an Albers Equal Area projection (ESRI:102008).

4. Fig. 4- top map: I don't think it necessary to show an "empty" map?
We included the top map of Figure 4 to indicate that, for the CDL, coverage of the reference data is consistently high. This information is necessary to understand why the third, combined map has some areas where coverage of reference data increases relative to the NVC. The values on the top map vary within the 93-100% category we used for visualization, so the map appears empty. If the reviewer has suggestions for how to visualize the drastically different range of values for CDL vs. NVC, while maintaining few enough categories for users to differentiate, we are happy to adjust.

5. The introduction should include a short synopsis of similar data integration efforts, to illustrate that this is a trending topic in general, and across disciplines. For example, a similar effort from the field of human settlement modelling would be this:
https://www.tandfonline.com/doi/full/10.1080/17538947.2018.1550121
We thank the reviewer for this feedback, which we addressed by adding the following text to the introduction:
"In addition to our contributions to land cover and environmental science, this work supports a broader scientific movement towards data integration, sharing, and re-use (The National Research Council 2010). Due to the rapidly increasing and large volume of available data, researchers are increasingly modifying and integrating existing datasets rather than creating entirely new data. From limnology (Soranno et al., 2015), human settlement mapping (Florczyk et al., 2020), environmental health (Heacock et al., 2022; Cui et al., 2022), paleoanthropology (Reed et al., 2018) to macroecology (Fletcher et al., 2019), to name a few, scientists in many disciplines are engaged in similar efforts to standardize and integrate geospatial datasets."